# Terminomics Methodologies and the Completeness of Reductive Dimethylation: A Meta-Analysis of Publicly Available Datasets

**DOI:** 10.3390/proteomes7020011

**Published:** 2019-03-29

**Authors:** Mariella Hurtado Silva, Iain J. Berry, Natalie Strange, Steven P. Djordjevic, Matthew P. Padula

**Affiliations:** 1Proteomics Core Facility and School of Life Sciences, Faculty of Science, University of Technology Sydney, Broadway NSW 2007, Australia; Mariella.I.Hurtado@alumni.uts.edu.au (M.H.S.); iain.berry@uts.edu.au (I.J.B.); natalie.strange@student.uts.edu.au (N.S.); 2The ithree Institute, Faculty of Science, University of Technology Sydney, Broadway NSW 2007, Australia; steven.djordjevic@uts.edu.au

**Keywords:** terminomics, mass spectrometry, amine labelling

## Abstract

Methods for analyzing the terminal sequences of proteins have been refined over the previous decade; however, few studies have evaluated the quality of the data that have been produced from those methodologies. While performing global N-terminal labelling on bacteria, we observed that the labelling was not complete and investigated whether this was a common occurrence. We assessed the completeness of labelling in a selection of existing, publicly available N-terminomics datasets and empirically determined that amine-based labelling chemistry does not achieve complete labelling and potentially has issues with labelling amine groups at sequence-specific residues. This finding led us to conduct a thorough review of the historical literature that showed that this is not an unexpected finding, with numerous publications reporting incomplete labelling. These findings have implications for the quantitation of N-terminal peptides and the biological interpretations of these data.

## 1. Introduction

Quantitative proteomics has become a widely used tool in the biological sciences, inferring biological significance from the changes in protein abundance between varying treatments or conditions. Protein quantification studies normally use an organism’s reference genome to identify and quantify products of an organism’s open reading frames (ORFs). This approach, while useful for quantitation of predicted protein products, does not take into consideration the various post-translational modifications (PTMs) that may occur on a protein, or the presence of biologically distinct proteoforms arising from a single ORF. Identifying specific proteoforms and quantifying their abundance levels is necessary for a complete assessment of proteomic variation and its biological significance.

The production of mature proteoforms is an intricate process, as outlined in Figure 1 [1,2,3]. ORFs by definition are inferred by bioinformatic prediction from genome sequencing projects, often without direct proteomic evidence to confirm their accuracy [4,5,6]. Variations resulting from chemical modifications to the nascent protein (PTMs), such as acetylation and phosphorylation, as well as primary structural modifications by proteolytic cleavage, introduces a level of proteome complexity that is often overlooked when using reference genomes to detect the presence of a particular ORF using mass spectrometry [7,8,9]. Considering proteolytic cleavage in particular, the most prevalent PTM in biological systems, mature proteoforms that are products of proteolysis are often incorrectly represented in protein databases as a non-mature, direct translation of the ORF. Our extensive work examining *Mycoplasma spp*., considered the ‘simplest’ self-replicating organism yet discovered, has shown that proteolytic cleavage is a critical process in the generation of mature proteoforms from large ORFs, producing a larger proteome than bioinformatically predicted for these genome-reduced bacteria [10]. In addition, proteolysis creates proteoforms that have different functionality to the parent proteoform, further extending proteome complexity [10,11,12,13,14,15,16,17,18,19,20,21,22]. This increase in proteome diversity through proteolysis also occurs in eukaryotes, and this presents a need to identify and characterize proteoforms produced through proteolysis in order to understand proteoform diversity and its effect on biological systems, rather than quantifying the abundance of aggregated ORF products. However, despite a range of methodologies being available to achieve this, the methods are often not able to definitively identify the diversity of proteoforms on a proteome-wide scale or in a high throughput manner.

The original method for analyzing intact proteoforms and protein sequences was Edman degradation, which allows direct and unambiguous sequencing of the N-terminal amino acids of intact, purified proteins [3,26,27,28]. Nevertheless, Edman degradation is a time-consuming process, sequencing one amino acid residue per hour, and is limited by the efficiency of the chemical reagents to hydrolyze the peptide bonds of each subsequent residue [26,27,28,29]. The ideal solution for characterizing intact proteoforms is direct sequencing in a complex proteome sample using intact mass spectrometry (MS), with high-resolution accurate mass measurements and MS/MS fragmentation. Intact protein MS/MS (or Top Down MS) analysis can determine the N-and C-terminal sequences of a proteoform by matching of spectral data to the predicted ORF, as well as detecting the presence and location of PTMs, where the difference in proteoform mass corresponds exactly to the PTM mass [30,31,32]. However, higher throughput and increased proteome depth (at the expense of proteoform identification) can be achieved by analyzing peptides from enzymatically digested proteins using traditional shotgun Liquid Chromatography-Tandem Mass Spectrometry (LC/MS/MS), with routine identification of ~11,000 ORF products [33,34,35] compared to <1500 proteoforms identified by Top Down MS [36]. In order to characterize the N- or C-terminus of the mature proteoforms, the terminal peptide must be unambiguously identified to avoid conjecture, which can be challenging if the true terminal peptide is not ‘MS friendly’ and thus undetectable due to poor ionization potential or unsuitable length after digestion. A solution for these issues which has been implemented by several groups [37,38,39,40] is the application of protein-level labelling and enrichment of terminal peptide sequences after proteolytic digestion, known collectively as “terminomics”. While these methods are able to reduce the aforementioned ambiguity, they are not free of problems.

Currently, the most popular terminomics techniques target the identification of the N-terminus of proteoforms by enriching and sequencing N-terminal peptides by MS, which are then mapped to ORFs in silico [41,42,43]. Enrichment aims to address the biggest obstacle encountered in terminomics and proteomics, which is the complexity of the sample being analyzed, as a sample containing tens of thousands of proteoforms will produce an exponentially larger number of peptides of varying abundance to be analyzed following enzymatic cleavage [44,45]. This is exemplified by the fact that the 15,721 human proteins or ORFs that have been detected and catalogued in ProteomicsDB, arguably the most comprehensive proteome resource available, are described by 455,289 unique peptides, with peptide evidence for 7977 N-termini and 6778 C-termini [46]. The competition of peptides for detection in MS can be addressed through the selected isolation of only N-terminal peptides via N-terminomics enrichment strategies, enabling the complexity of the sample to be reduced and minimizing signal competition during MS, thereby improving N-terminal sequence identification [41].

Bottom-up MS proteome analysis assigns peptide sequences to ORFs using bioinformatics; however, information about the original intact proteoform is often lost [3,47,48]. For example, proteoforms differing by a single amino substitution cannot be distinguished if the peptide containing the substitution is not detected. In the case of proteolytic processing, the main problem of bottom-up methodologies is the previously mentioned unambiguous assignment of the N-termini of mature proteoforms. This is best illustrated by considering a digest of Bovine Serum Albumin (BSA). In a majority of cases when analyzing peptides, the MS^1^ scan range is usually set between 350–1500 m/z to optimize the transmission of ions through the quadrupole for high sensitivity and to avoid low mass, but high signal ‘background’ ions being detected. The lower mass of 350 m/z is also set to avoid selecting small peptides of less than 3–5 amino acids (AAs) for fragmentation and MS/MS, as these are scored against by search algorithms as they are more likely to be matched randomly. In the case of BSA, the N-terminal site is well known (UniProt P02796) and the mature proteoform starts at Aspartic Acid (D, position 19) after removal of the signal and pro-peptide. After digestion with trypsin, the peptide created at the N-terminal of the proteoform is DTHK, which would have a monoisotopic mass of 500.24 m/z, while the 2^+^ ion would be 250.62 m/z, which is below the normally utilized scan range. The first detectable and assigned peptide of the sequence in our experience is FKDLGEEHFK (AAs 34–44) which includes a missed tryptic cleavage site. If this was an uncharacterized protein with no knowledge of the mature proteoform, such shotgun LC/MS/MS data would erroneously assign the N-terminal amino acid.

The difficulty of identifying proteoforms using shotgun LC/MS/MS is further compounded when trying to characterize proteoforms generated by the cleavage of large precursor proteins, exemplified by the adhesin families of *Mycoplasma hyopneumoniae*. This organism overcomes a small genome of less than 700 ORFs by performing extensive proteolytic cleavage of expressed proteins into mature proteoforms, such as the cleavage of mhp683 [12]. This ORF is processed into three main proteoforms of 45, 48, and 50 kDa, a fact not able to be determined by shotgun LC/MS/MS but revealed through prior fractionation by proteoform mass using one dimensional PolyAcrylamide Gel Electrophoresis (SDS-PAGE), where peptides mapping to the entire ORF are found at approximately 50 kDa. The individual proteoforms are resolvable by Two Dimensional PAGE (Isoelectric focusing followed by SDS-PAGE), and LC/MS/MS analysis of peptides from the 48 kDa proteoform (designated P48) revealed the true N-terminus due to the presence of pyroglutamate, formed by the cyclisation of the side chain of glutamate or glutamine with its free α-amine. This ‘labelling’ of the N-terminus, where the mass of the AA has been altered by the formation of pyroglutamate (which can be removed by enzymatic treatment with pyroglutaminase), led us to explore the literature for other methods of labelling to distinguish the true N-termini of mature proteoforms. Most strategies for N-terminomics, such as COmbined FRActional DIagonal Chromatography (COFRADIC) and Terminal Amine Isotopic Labeling of Substrates (TAILS), implement the use of a chemical label [42,49,50,51,52], where sample complexity is then simplified by either positive or negative enrichment of labelled N-terminal peptides for analysis by LC/MS/MS [3].

The most popular approach to N-terminal labelling of proteoforms involves exploiting the apparent chemical reactivity of the free primary amine group on the N-terminus and lysine side chain [3,43,53]. As is the case for numerous chemical reactions, including Edman degradation, labelling procedures are restrained by the efficiency of the chemical reaction to modify the targeted amine groups and attach the chemical label [40,43,54]. In our laboratory, there has been evidence to suggest incomplete dimethyl labelling of proteoforms isolated from bacteria, especially the lysine-rich Mycoplasmas, with cases of lysine being identified as the N-terminal amino acid but possessing only one dimethyl tag instead of the expected two (Figure 2). This observation was made because our peptide search parameters have dimethylation set as a variable modification, which is in contrast to the consensus in the literature to use fixed modifications. This led us to suspect that incomplete labelling was more widespread than being reported. Several groups have explored the efficiency of the dimethylation procedure (listed in Table 1), but each quote different sets of conditions that are “optimal” for protein sample labelling. Additionally, none of these groups have systematically reported the effect of protein concentration or the level of proteome complexity that is compatible with the dimethyl–amine chemistry or other labelling techniques [55,56,57].

Examination of data from experiments using reductive dimethylation in our laboratory and comparing it to other reports prompted us to question whether the dimethylation labelling was complete with all primary amines labelled when performed on different organisms with higher lysine content. This also raised the question of whether the data in current N-terminomics literature is underreporting the completeness of labelling, potentially leading to inaccurate quantitation. To address this, we have performed a meta-analysis on a selection of publically available, quantitative datasets which implemented a dimethyl labelling protocol without peptide depletion.

## 2. Materials and Methods

### 2.1. Data Selection

Raw MS/MS data files were obtained from the online proteomics data repository, PRoteomics IDEntifications (PRIDE) Archive (Table 2). All analyzed datasets implemented dimethyl labelling protocols involving no enrichment strategy and labelling at the peptide level (after sample digestion); thus, all detected peptides should be dimethylated if all primary amines had reacted. Our meta-analysis omits datasets with ambiguously named data files, no raw MS/MS data, or labelling strategies which utilized peptide depletion protocols (e.g., N-TAILS). The reason for omitting datasets generated with depletion strategies is that the negative selection of amine-containing peptides, which should be the internal peptides generated by proteolytic digestion after dimethylation, will remove any incompletely labelled peptides, preventing their analysis by LC/MS/MS. We were therefore forced to omit studies that used dimethylation at the protein level, the technique that is of most value for the determination of the N-termini of mature proteoforms, because the relevant data was not being captured in the experiment. The datasets used for re-analysis were selected as randomly as possible to cover as many types of organisms as possible.

### 2.2. Data Search and Analysis

Data files obtained from PRIDE were subsequently searched using the PEAKS Studio software package (v8.5) with the relevant organism sequence database, while the specific instrument parameters were as per those reported in each study, but with the relevant modifications set to variable rather than fixed (see Appendix A for full specifications). The PEAKS search results were filtered to remove sample contaminants and then sorted to identify spectra that match to the same sequence (duplicate sequences). All significant (*p* ≤ 0.05) duplicate sequences were interrogated for frequency of complete, partial, or total lack of dimethyl labelling by assessing the proportion of duplicate sequences detected with a label at the N-terminus and at any lysine residues.

## 3. Results

In this meta-analysis, we sought to examine whether dimethylation was present on all primary amines in peptides in published data utilizing a non-depleted dimethyl labelling protocol. It is important to point out that many of the studies examined performed database searching with dimethylation as a fixed modification, thus assuming that all primary amines present were dimethylated. If dimethylation of all available reactive amines does not occur, as we suspect that it does not, the search parameters applied in these studies will either not assign all of the acquired spectra, resulting in a false negative, or assign a spectrum to the incorrect sequence, resulting in a false positive. Our analysis, presented in Table 2, found that between 6–18% of duplicate sequences detected in each dataset have amine groups that are not dimethylated, which was not reported by the authors. These peptide matches are therefore false negatives, a value that is normally not able to be calculated. As these are peptide-based, shotgun LC/MS/MS experiments, the need to calculate false discovery rates (FDR) is a mandatory requirement of many reviewers and journals, and it is generally accepted that the FDR be below 1% [58,59,60]. Our analysis assigned FDRs to each dataset of ≤3%. In the case of these studies, the false negative rate is far greater than the false positive rate reported in the individual studies, and this is of serious concern when quantitation is considered as it will lead to false values.

The meta-analysis also revealed that a portion of the acquired MS/MS spectra are unassigned in the original publications, indicating that potentially important protein/peptide information is being overlooked in the final quantification. The number of peptides displaying a dimethyl label varied between each sample, with only 81–94% of all peptides demonstrating a dimethylated -amine and, in the case of lysine terminating peptides, a dimethylated -amine. It is possible that this is an underestimation. Prior published evidence [61,62] indicates that the ability of a primary amine to be dimethyl-labelled may be relative to the characteristics of the protein sample (reviewed in greater detail by Feeney and Blankenhorn [63]). This data may suggest that there are variations in labelling between organisms; however, this view is not supported by the meta-analysis conducted here. Jentoft and Dearborn [61] provided evidence that extreme concentrations of formaldehyde and NaBH_3_CN will not result in complete modification of primary amines in proteins, but concluded that reductive methylation may be implemented quantitatively. In contrast, Gidley and Sanders [64] found that the yields are never quantitative, and there always appears to be unchanged starting material present at the completion of the reaction, which is in contrast to the current understanding and implementation of the technique as reported in the literature.

One study that does report a completeness of labelling is Rowland et al. [65], where the labelling efficiency of lysine residues in the chloroplast proteome of A. thaliana was reported to be ≥99% for detected peptides. This may be attributed to protein preparation and reaction conditions, or the inherent characteristics of proteoforms present in the investigated proteome [61,62,66]. While these data [65] are available on the online proteomics data repository PRIDE (dataset identifier PXD002476 and 10.6019/PXD002476), it was difficult for us to determine with complete certainty which files corresponded to the dimethyl efficiency testing, so the data was unable to be included in our meta-analysis. This is not an uncommon issue, and there needs to be a dedicated effort made to properly label raw data files so that the results can be independently validated.

Inconsistency in labelling of raw data files available from online data repositories was only one issue encountered that hindered data acquisition. A large number of studies performing reductive dimethylation experiments failed to upload raw data files onto online data repositories, which restricted the number of datasets available for analysis. Results from our meta-analysis indicate that in the datasets analyzed, dimethylation was not complete; however, more data is required to understand the more widespread implications of this observed inefficiency. As such, we acknowledge and echo the recommendations of Lange et al. [67] and strongly encourage others to upload raw data files with clear, understandable filenames onto online data repositories.

In our meta-analysis, we were unable to include datasets from studies utilizing TAILS or negative selection of peptides because incompletely labelled peptides would be captured by the polyaldehyde polymers used to capture all molecules with free primary amines. These studies implement dimethylation at the protein level, which is of most relevance to our need to identify mature proteoforms. In our experience, protein level labelling is not complete, and we have no reason to suspect that protein level labelling is complete in TAILS-based analysis or other systems. Studies implementing chemical labelling strategies for quantitative analyses need to be aware that labelling efficiency can be variable and incomplete with important implications for data analysis, so we suggest that the extent of this should be empirically analyzed by searching the data with demethylation as a variable modification to determine the completeness of labelling. Once determined, the researcher can decide whether to proceed with the quantitative experiment and report the completeness of amine labelling. In SILAC experiments, it is generally accepted that the heavy amino acid be incorporated into >95% of the peptides detected before the quantitative experiment is performed, and this should be the case for chemical labelling strategies.

## 4. Discussion

Over a number of years, our laboratory has been interested in the characterization of proteolytic processing that occurs in prokaryotes to generate proteome diversity from a relatively small genome. Through the use of protein-centric techniques, especially 2D-PAGE, we showed the extent of processing in the model bacteria *Mycoplasma hyopneumoniae*, but we were unable to definitively identify the point of cleavage because we could not be sure that a peptide closer to the N-terminal was not being detected because it was not ‘MS-friendly’. In an attempt to resolve this, we turned to reductive dimethylation as a method of labelling the N-terminal amine of mature proteoforms, but we found that the ‘completeness’ of the labelling was less than that reported in the wider literature. In an attempt to understand why this was the case here, we have performed a meta-analysis on a random selection of N-terminomics datasets with surprising results. The significant issue is that the search parameters used to identify dimethylated peptides assume that all amines are modified, which is a flawed assumption as very few chemical reactions proceed to absolute completion and some potential reactants are always left over. During reanalysis, we found a significant number of unlabeled peptides in the datasets, which are false negatives in the original published analysis. This brings into question some of the conclusions of any publications seeking to use reductive dimethylation in a quantitative manner. It is clear that more work needs to be performed to further characterize the reductive dimethylation chemistry and workflows. During our analysis, we had difficulty identifying datasets which met the selection criteria for the meta-analysis, due to poor annotation of datasets in PRIDE or the use of enrichment tools which disguise the presence of unlabeled peptides. It is clear that we need to investigate other chemistries, such as succinimide-based chemistries utilized in Isobaric Tags for Relative and Absolute Quantitation (iTRAQ) or Tandem Mass Tags (TMT) protocols, which may provide a more complete labelling or may suffer similar shortfalls as the dimethylation reaction.

## Figures and Tables

**Figure 1 proteomes-07-00011-f001:**
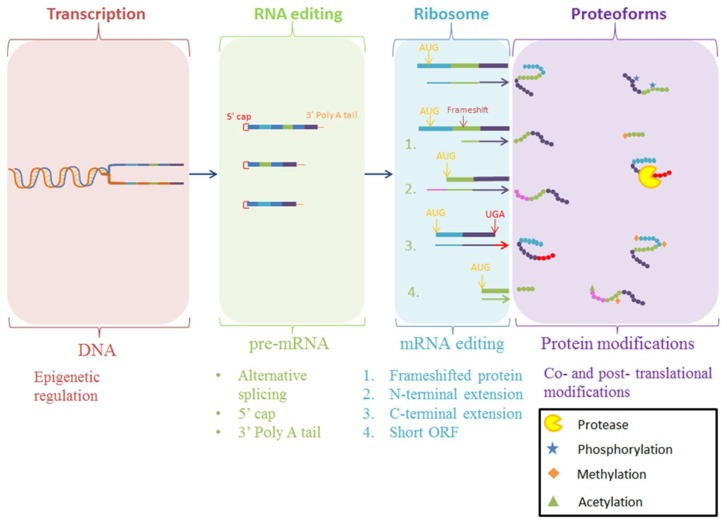
Schematic for the various methods for the production of proteoforms. Final protein products may be modified during transcription or during translation at the ribosome. Alternatively, nascent polypeptide chains may be modified after translation with a myriad of post-translational modifications. Once translated, the protein sequence can often require further modifications to perform a specialized function, generating proteoforms that vary from the original expressed protein [2,3,23]. Post-translational modifications such as phosphorylation, a reversible chemical addition to the protein, or proteolysis, a permanent hydrolysis event removing amino acid(s) from the polypeptide chain, cause important functional changes in proteins [24,25].

**Figure 2 proteomes-07-00011-f002:**
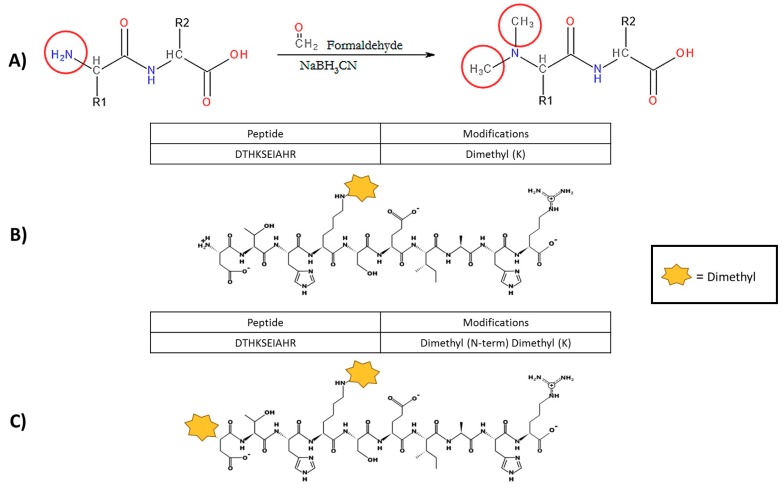
Dimethylation reaction implemented in the UTS Proteomics Core Facility with examples of incomplete labelling. (**A**) Theoretical reductive dimethylation reaction shown, which will attach two methyl groups to every prime amine in a protein sample (N-terminus and lysine residues). However, preliminary experimental results of the dimethylation process indicate that the current method is not modifying all primary amine groups in complex samples. (**B**) Unpublished data of incomplete labelling occurring in a model protein sample (bovine serum albumin). The N-terminal sequence that was obtained indicated that the protein amino acid sequence began with aspartic acid (Letter symbol, D), which was confirmed with bovine serum albumin data from UniProt (accession number: P02769). Identical peptide sequences containing lysine residues have been sequenced by mass spectrometry; however, the first peptide, indicated in the red rectangle, has been identified with only the lysine dimethylated. (**C**) The second sequence from the same mass spectrometry experiment has a dimethyl label on both the N-terminal amine (the aspartic acid residue) and lysine residue.

**Table 1 proteomes-07-00011-t001:** Published reaction conditions of reductive methylation protocols.

Reference (Year)	Reaction Conditions	Reactant Concentrations	Significant Observations
Friedman et al. [66] (1974)	4–16 h (room temperature) Alcohol/lithium acetate buffer pH 5.2	~11 mM NaBH_3_CN ~11 mM aldehyde (various)	Modification of lysine resides ranged from 40–90% using different aldehyde reagents, between protein molecules and different amino acid residues
Jentoft et al. [61] (1979)	2–24 h (22 °C) HEPES buffer pH 7.5	20 mM NaBH_3_CN Concentration formaldehyde ~ concentration of lysyl residues in sample	80–90% dimethyl conversion of lysine residues with a 6 fold excess of formaldehyde Lower concentrations of NaBH_3_CN (5 mM to 20 mM) yielded in the highest modifications of lysyl residues Maximal rates of labelling observed at pH 8
Hsu et al. [68] (2005)	Sodium acetate buffer pH 5–85 min	~22 mM NaBH_3_CN ~52 mM formaldehyde	Observation of immonium ion signal with dimethyl labelling
Krusemark et al. [69] (2008)	2 h, room temperature 300 mM triethanolamine and 6 mM Guanidine–HCL buffer pH 7.5 20% MeOH 1 mg/mL protein	30 mM Pyridine-BH_3_ (reducing agent) 20 mM formaldehyde	4 model proteins containing various abundance of amine groups, dimethyl labelled to completeness NaBH_3_CN and NaBH_4_ found to produce side reactions resulting in reduced purity of products
Boersema et al. [70] (2009)	1 h, room temperature 100 mM Triethylamonium bicarbonate buffer pH 5–8.5	~22 mM NaBH_3_CN ~52 mM formaldehyde	(protocol paper)
Kleifeld et al. [43] (2011)	4 h—overnight incubation at 37 °C 100 mM HEPES pH 7.0	20 mM NaBH_3_CN 40 mM formaldehyde	(protocol paper)
Jhan et al. [71] (2017)	30 s–2h, room temperature 100 mM sodium acetate pH 5–6	1.4–85 mM NaBH_3_CN 156 mM formaldehyde	Accessibility of primary amines on the protein greatly affects dimethylation efficiency At 30 s 80% of amines were dimethylated

**Table 2 proteomes-07-00011-t002:** Meta-analysis results of dimethyl labelling studies [72,73,74,75].

PRIDE Dataset Identifier	FDR PEAKS Generated (%)	Duplicate Peptide Sequences Detected	Duplicate Sequences with Complete Labelling	Complete Labelling (%)	Duplicate Sequences with Partial Labelling	Partial Labelling (%)	Duplicate Sequences with No Dimethyl Label	Unlabeled (%)	Total Partial and Unlabeled Duplicate Sequences	Total Partial and Unlabeled (%)
PXD002785 PXD003833 (125)	1.7	6658	5454	81.92	1161	17.44	43	0.65	1204	18.08
PRD000055 (115)	0.6	5395	5062	93.83	315	5.84	18	0.33	333	6.17
PXD005920 (126)	1.5	3269	2847	87.09	404	12.36	18	0.55	422	12.91
PXD003298 (127)	1.6	3531	2893	81.93	584	16.54	54	1.53	638	18.07
PXD004654 (128)	3.0	6293	5498	87.37	715	11.36	80	1.27	795	12.63

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
