# Peer review of "Terminomics Methodologies and the Completeness of Reductive Dimethylation: A Meta-Analysis of Publicly Available Datasets"

_proteomes, 2019, doi:10.3390/proteomes7020011_

Round 1
Reviewer 1 Report
This manuscript presents meta-analysis of publically available data sets for terminal sequences that utilized reductive demethylation labeling. There is increasing emphasis for the re-use of public data sets but there are problems to re-analyze and utilize cross-lab data due to lack of uniformity in sample preparation and lack of standardized representation. Efforts by the authors to re-visit public data for re-evaluation and extract new information is exciting and much appreciated. N-terminal peptide labeling methods have improved over the years but still are far from being perfect and still there are issues with labeling efficiency and data quality.
Major points:
The introduction is too long, and often redundant. It can be shortened to almost half. Focus more on issues related to labeling N-terminal peptides in general and need for re-evaluation of pubic data sets for accuracy and limitation, etc.
In contrast, method section is too limited.
2.1. Data selection: Raw MS/MS data files were obtained from the online proteomics data repository, PRIDE Archive (Table 1). The table 1 shows references from 1974-2017. Were there MS/MS data in Pride from Friedman, Williams [ref 63], and used for this study. The analytical and MS technology has advanced so much during the last 10 years, so it is not fair to compare data from 1974, 1979, 2005 to 2017. I could not understand why more recent data only were not considered. I might be missing something here.
There is no explanation how authors implemented data selection criteria. This needs to be clarified and justified. If 1974 and 1979 data needs to be included, I want to see the explanation. Why not compare data from 2017-2019, data collected in the same or similar instruments but using different labeling techniques??
I think it would be better to provide details of these public data used in this study (organisms, sample preparation, MS instruments, etc.) would be better. I could not find any supplementary files for those peptides/proteins. The only supplementary file is a table that shows PRIDE datasets used.
Line 215: The statement "the meta-analysis also revealed that a portion of the acquired MS/MS spectra are unassigned in the original publications, indicating that potential important protein/peptide information is being overlooked in the final quantification” I have no idea what authors are referring here (which publication, which peptide/protein). This is vague explanation and should be accompanied by actual example.
Line 229: One study that does report a completeness of labeling is Rowland et al., [62]. This data is not listed in table 1.
Overall, I this the data will be a beautiful resource for the community if presented with details and all the results made public. However, it falls a short of my expectation when it comes to explaining the methods, results and discussion of the observed labeled peptides/proteins. I hope authors will consider these comments and improve the manuscript quality
Author Response
Reviewer 1
This manuscript presents meta-analysis of publically available data sets for terminal sequences that utilized reductive demethylation labeling. There is increasing emphasis for the re-use of public data sets but there are problems to re-analyze and utilize cross-lab data due to lack of uniformity in sample preparation and lack of standardized representation. Efforts by the authors to re-visit public data for re-evaluation and extract new information is exciting and much appreciated. N-terminal peptide labeling methods have improved over the years but still are far from being perfect and still there are issues with labeling efficiency and data quality.
Major points:
The introduction is too long, and often redundant. It can be shortened to almost half. Focus more on issues related to labeling N-terminal peptides in general and need for re-evaluation of pubic data sets for accuracy and limitation, etc.
The reasoning for the length of the introduction is that we don’t believe that the need to properly identify the true N-terminal start site of a proteoform translated from a gene has been properly and simply explained. While the stated aim of our manuscript is to bring to light a problem, another aim is to provide an educational resource to novices in the field. Thus, we feel that the detail in the introduction is necessary.
In contrast, method section is too limited.
2.1. Data selection: Raw MS/MS data files were obtained from the online proteomics data repository, PRIDE Archive (Table 1). The table 1 shows references from 1974-2017. Were there MS/MS data in Pride from Friedman, Williams [ref 63], and used for this study. The analytical and MS technology has advanced so much during the last 10 years, so it is not fair to compare data from 1974, 1979, 2005 to 2017. I could not understand why more recent data only were not considered. I might be missing something here.
In the text, we mislabelled the table we wanted to refer to and this has been corrected. The references in Table 1 detail original articles that describe the reaction chemistry of reductive dimethyaltion rather than the application of the chemistry to proteomics experiments. Reference 63 dates from 1974 and mass spectrometry was not ultilised to generate the data. Therefore, there is no data in PRIDE from this publication. The oldest of the datasets we have used for our analysis dates to 2008. The majority of the PRIDE datasets are from 2014-2017. We expect to see similar problems in any similar dataset available because the problem is not in the acquisition of the data but in the processing of the data which leads to misleading conclusions. Using datasets generated on more sensitive instrumentation would likely reveal the same proportion of unlabelled peptides in a greater number of peptides.
There is no explanation how authors implemented data selection criteria. This needs to be clarified and justified. If 1974 and 1979 data needs to be included, I want to see the explanation. Why not compare data from 2017-2019, data collected in the same or similar instruments but using different labeling techniques??
This is described in section 2.1 of the manuscript. We have added the sentence ‘The datasets used for re-analysis were selected as randomly as possible to cover as many types of organisms as possible’. We tried not to impose any bias on the selection of the datasets because we did not want to be perceived as engaging in a ‘witch hunt’ for want of a better term.
I think it would be better to provide details of these public data used in this study (organisms, sample preparation, MS instruments, etc.) would be better. I could not find any supplementary files for those peptides/proteins. The only supplementary file is a table that shows PRIDE datasets used.
This is described in the manuscript. The PRIDE accession numbers for the datasets are listed in table 2 which can be used to find the information about organisms, sample preparation and instrumentation. It was not necessary to include this information in the manuscript because our observations and conclusions are independent of this information. We did not want this manuscript to be a ‘name and shame’.
Line 215: The statement "the meta-analysis also revealed that a portion of the acquired MS/MS spectra are unassigned in the original publications, indicating that potential important protein/peptide information is being overlooked in the final quantification” I have no idea what authors are referring here (which publication, which peptide/protein). This is vague explanation and should be accompanied by actual example.
The observation was made in all datasets and is detailed in table 2 which shows the percentage of unlabelled peptides that are identified when appropriate search settings are applied. We don’t see the value of including a specific example because there are 18-80 unlabelled peptides across the datasets and 315-1161 partially labelled peptides across the datasets, as listed in table 2.
Line 229: One study that does report a completeness of labeling is Rowland et al., [62]. This data is not listed in table 1.
This data was excluded from the analysis because it did not meet our criteria. As stated in the manuscript ‘While these data [62] are available on the online proteomics data repository PRIDE (dataset identifier PXD002476 and 10.6019/PXD002476), it was difficult for us to determine with complete certainty which files corresponded to the dimethyl efficiency testing, so the data was unable to be included in our meta-analysis.’ We did not contact the authors to resolve the poor labelling of their data because we did not need to use their data in our analysis.
Overall, I this the data will be a beautiful resource for the community if presented with details and all the results made public. However, it falls a short of my expectation when it comes to explaining the methods, results and discussion of the observed labeled peptides/proteins. I hope authors will consider these comments and improve the manuscript quality.
We thank the reviewer for their kind comment. We have amended some sections of the manuscript to make it clearer. The PRIDE numbers of the datasets used are listed in table 2. We did not want this article to become a ‘name and shame’ exercise and deliberately refrained from providing more details of the study as the problem we are pointing out is more widespread than just those studies analysed. This article strives to change data reporting practices for all articles of this type.
Reviewer 2 Report
This meta-analysis of primary amine reductive methylation is a well-written and useful synopsis and quantitative analysis of several published studies in this area. The results, while not unexpected, highlight the importance of not assuming quantitative conversion and raise questions regarding methodological improvement.
A few points for the authors:
(l130) A mention of using pyroglutaminase to deblock N-terminal pyroglutamine should be included.
(Figure 2) Reformat the boxed text using a plain text font, and avoid reproducing software outputs. Remove any unnecessary data and explain remaining terms.
(l195) Specify the parameters used by the inhouse script to quantify labelling efficiency.
(Table 2) Clarify whether any of these studies correspond to any of the studies listed in Table 1.It would also be useful to know the labelling conditions used in these studies, possibly based on those listed in Table 1.
(l229 and/or Discussion) The role of protein folding in inhibiting modification, and denaturation as a means in improving the level of protein modification should be addressed.
Author Response
Reviewer 2.
This meta-analysis of primary amine reductive methylation is a well-written and useful synopsis and quantitative analysis of several published studies in this area. The results, while not unexpected, highlight the importance of not assuming quantitative conversion and raise questions regarding methodological improvement.
A few points for the authors:
(l130) A mention of using pyroglutaminase to deblock N-terminal pyroglutamine should be included.
We have included a mention about enzymatic removal of pyroglutamate.
(Figure 2) Reformat the boxed text using a plain text font, and avoid reproducing software outputs. Remove any unnecessary data and explain remaining terms.
Revised in manuscript.
(l195) Specify the parameters used by the inhouse script to quantify labelling efficiency.
This has been added to the manuscript.
(Table 2) Clarify whether any of these studies correspond to any of the studies listed in Table 1.It would also be useful to know the labelling conditions used in these studies, possibly based on those listed in Table 1.
Of the references in table 1, only Boersema et al 2009/PRD000055 was further analysed because much later work uses this method for their labelling. The labelling conditions are listed in the information provided on PRIDE and in the articles referenced there.
(l229 and/or Discussion) The role of protein folding in inhibiting modification, and denaturation as a means in improving the level of protein modification should be addressed.
Our own work with Mycoplasma has been performed under denaturing conditions and we observe a lack of complete labelling. In saying this, there is no systematic evaluation of the effects of a wide variety of reaction conditions on labelling efficiency. We consider this to be essential future work.
Reviewer 3 Report
The manuscript “Terminomics methodologies and the efficacy of reductive dimethylation: a meta-analysis of publically available datasets” by Silva et al. touches an important subject in proteomics. A myriad of proteoforms exist in cells of which a substantial part is generated via proteolytical cleavage. Moreover, it is essential for biological interpretation to know wether or not proteins have been processed by proteases and if these species are up- or down-regulated in certain conditions as only specific proteoforms might represent functional protein variants. Top-down proteomics would be the method of choice to map and quantify exact proteoforms. However, as the authors indicate in the manuscript, this approach is limited in throughput and provides smaller proteome coverage compared to bottom-up approaches. Terminomics provides a partial solution as it enables specific detection of proteoforms with differential N-termini. However, this methodology relies on chemical labeling of the N-terminus prior to enzymatic digestion during sample preparation which requires the N-terminal labeling to be specific, efficiënt, and reproducible. The authors focus on the dimethyl labelling approach which is commonly used in shotgun proteomics. Database search results with their own data as well as publically available datasets identified both labelled and non-labeled N-terminal peptides which triggered the work described in their manuscript. Incomplete N-terminal labelling is an important topic to address in N-terminomics which definitely requires more attention. The intention of the work described in the manuscript by Silva et al is sound but I do have major concerns that I hope the authors are willing to address by additional analyses and according adaptation of the statements and conclusions in the manuscript.
My main concerns are about the data processing and subsequent analysis as performed. Database searches were performed using dimethyl labelling as variable modification to identify both labelled and unlabelled versions of peptides in samples that were labelled after tryptic digestion. Using this approach, the authors found that partial labelling was detected for 5.84-17.44% of the unique peptide sequences within these datasets. Based on this results the authors state that “This brings into question some of the conclusion of any publications seeking to use reductive dimethylation in a quantitative manner”. However, analysis of datasets based on exclusively database search results without taking quantitative information into account likely presents a biased picture. One should take into account that it is nearly impossible to achieve 100% labelling efficiency and specificity. Nevertheless, chemical labelling can be used for quanitative purposes if the labelling reaction achieves high enough efficiency (>95% overall efficiency) and good reproducibility. Database search results by itself do not provide the required information to assess the labelling efficiency. For example, if 99% of a particular peptide would be labelled and has 1E6 precursor intensity this means that the unlabelled counterpart might be detected at 1E4 intensity which is sufficient for successful fragmentation experiments and subsequent database search identification. This is e.g. also commonly observed for carbamidomethylated cysteines containing high abundant peptides in shotgun proteomics. Although the modified peptide is perfectly useable for quantitative purposes the database search result detects it as “partially labelled”. Database search results might thus “indicate” labelling issues but readily underestimate the true labelling efficiency. In other words: “The labelling efficiency of duplicate peptides was analyzed using an in-house script” is an overstatement (in addition: what script language, what did it analyze exactly, how was its output validated?).
It is thus important to use quantitative information next to database search results which has not been performed in this study. There are several possibilities available to the authors to explore the labelling efficiency of which a few are mentioned here. A first step would be to calculate the ratio between the labelled and unlabelled version(s) of each peptide sequence using non-normalized signal intensities. It is true that retention shifts due to the label or the label itself might influence the signal intensity but this is non-significant for dimethy labelling at a global level in our experience. Further, if incomplete labelling does interfere with quantitation this should be obvious from the relative quantitation results: the authors could test if the ratio between the labelled pairs statistically differ from the ratios of the other peptides from the same protein in these datasets, particularly if label swaps were performed. If the labelling efficiency is consistent between samples then the partially labelled peptides will (globally) show a similar distribution of respective ratio deviations as the complete labelled peptides and will not impact quantitative results. There are more alternative analyses feasible to investigate IF the labelling efficiency is indeed low in these datasets and IF the lower labelling efficiency impacts quantitative results (hence: increased variation should be substantially larger than the intra- and inter-assay variation; proper statistical analysis is warranted in any case).
Minor concerns:
Figure 2: the authors show the best peptide-spectrum match (PSM) in a mascot search result from a (poor mass accuracy) dataset. The information shown in figure 2 does not proof the actual modification site by itself (particularly important for 2B). (1) provide delta scores for the subsequent lower (or equal scoring) peptide-spectrum matches for both spectra. (2) please provide the annotated MS/MS spectra with indicated diagnostic fragment ions to proof the site-specific localization of the dimethyl group.
General remark with respect to the database searches: N-terminal protein formylation might occur in specific species or organelles. We routinely detect N-terminal formylation of specific mitochondrial proteoforms in different organisms by top-down proteomics and subsequent shotgun proteomics. The small difference in the resulting mass increase of a dimethyl label (+28.0313 Da) and a formyl group (+27.994915 Da) might give problems in selected cases. Dimethylation might not be discerned from formylation for peptides of Mr >2500 Da at <15ppm precursor mass tolerance with near perfect mass calibrated data files. However, if there is a few ppm mass offset in the acquired data formylated peptides can be readily detected as dimethylated peptides and vice versa. Did the authors consider this potential problem and performed variable PTM searches to check for potential issues?, what were the precursor mass error distributions of the datasets and how were these re-calibrated (if at all)? In addition, what were the exact protein sequence databases used in this study (source, version/date?)?
Line 201-203: “... the search parameters applied in these studies will either not assign all of the acquired spectra, resulting in a false negative, or assign a spectrum to the incorrect sequence resulting in a false positive”. Shotgun proteomics will never achieve 100% peptide-spectrum matches for a multitude of reasons (a.o. poor signal-to-noise spectra, poor fragmentation, chimeric spectra, sequence scrambling, unspecified combinations of modifications and amino acid variants, incorrect protein sequences in the database, statistics, etc etc).
Line 211-212: “In the case of these studies, the false negative rate is far greater than the false positive rate reported in the individual studies and this is of serious concern when quantitation is considered as it will lead to false values”. The false negative rate is always (!) higher than the false positive rate in decoy database search strategies for any given shotgun proteomics experiment. However, this by itself will not lead to false quantitative values per se IF the labelling efficiency is nearly identical between both samples. Moreover, the authors did not determine the false negative rate – this can by definition not be calculated for these datasets since it lacks ground truth information. It can be estimated by alternative statistical approaches that require a lot of mathematics and that take more than just missing dimethyl labels into account.
Line 229-245: This section highlights the unfortunate state of many datasets filed at online repositories and stresses the need to adhere to FAIR guidelines. However, in our experience authors that deposited datasets are often willing to provide additional information or data when asked to. Did the authors of this manuscript contact the owners of these datasets and asked for information / data files?
Line 258-261: did the authors consider using different enzymes or chemical digestion methods to validate N-terminomics results / find missing N-termini? Why would N-terminal dimethylation solve the problem of “MS-unfriendly” N-terminal peptides? The dimethyl labelling confirms the N-terminus IF an N-terminal dimethylated peptide is detected but does not solve the problem of absent proof. A single enzyme will nearly certain not provide “MS-friendly” peptides for each N-terminus of every canonical protein sequence. As such, different digestion protocols can (or should) be used to fill missing data gaps and for cross-validation purposes. Moreover, calculations can be performed a priori to evaluate detectability of protein N-termini with the use of different digestion protocols. Moreover, it would be best to measure a sample before and after labelling of protein N-termini as PTM mass offsets in database searches can lead to ambiguous PTM identifications. In general, FDR calculations based on decoy database search strategies are just an estimate of the true false-discovery rate and individual incorrect PSMs will always occur even at extremely low FDR thresholds.
Author Response
Reviewer 3.
The manuscript “Terminomics methodologies and the efficacy of reductive dimethylation: a meta-analysis of publically available datasets” by Silva et al. touches an important subject in proteomics. A myriad of proteoforms exist in cells of which a substantial part is generated via proteolytical cleavage. Moreover, it is essential for biological interpretation to know wether or not proteins have been processed by proteases and if these species are up- or down-regulated in certain conditions as only specific proteoforms might represent functional protein variants. Top-down proteomics would be the method of choice to map and quantify exact proteoforms. However, as the authors indicate in the manuscript, this approach is limited in throughput and provides smaller proteome coverage compared to bottom-up approaches. Terminomics provides a partial solution as it enables specific detection of proteoforms with differential N-termini. However, this methodology relies on chemical labeling of the N-terminus prior to enzymatic digestion during sample preparation which requires the N-terminal labeling to be specific, efficiënt, and reproducible. The authors focus on the dimethyl labelling approach which is commonly used in shotgun proteomics. Database search results with their own data as well as publically available datasets identified both labelled and non-labeled N-terminal peptides which triggered the work described in their manuscript. Incomplete N-terminal labelling is an important topic to address in N-terminomics which definitely requires more attention. The intention of the work described in the manuscript by Silva et al is sound but I do have major concerns that I hope the authors are willing to address by additional analyses and according adaptation of the statements and conclusions in the manuscript.
My main concerns are about the data processing and subsequent analysis as performed. Database searches were performed using dimethyl labelling as variable modification to identify both labelled and unlabelled versions of peptides in samples that were labelled after tryptic digestion. Using this approach, the authors found that partial labelling was detected for 5.84-17.44% of the unique peptide sequences within these datasets. Based on this results the authors state that “This brings into question some of the conclusion of any publications seeking to use reductive dimethylation in a quantitative manner”. However, analysis of datasets based on exclusively database search results without taking quantitative information into account likely presents a biased picture. One should take into account that it is nearly impossible to achieve 100% labelling efficiency and specificity. Nevertheless, chemical labelling can be used for quanitative purposes if the labelling reaction achieves high enough efficiency (>95% overall efficiency) and good reproducibility. Database search results by itself do not provide the required information to assess the labelling efficiency. For example, if 99% of a particular peptide would be labelled and has 1E6 precursor intensity this means that the unlabelled counterpart might be detected at 1E4 intensity which is sufficient for successful fragmentation experiments and subsequent database search identification. This is e.g. also commonly observed for carbamidomethylated cysteines containing high abundant peptides in shotgun proteomics. Although the modified peptide is perfectly useable for quantitative purposes the database search result detects it as “partially labelled”. Database search results might thus “indicate” labelling issues but readily underestimate the true labelling efficiency. In other words: “The labelling efficiency of duplicate peptides was analyzed using an in-house script” is an overstatement (in addition: what script language, what did it analyze exactly, how was its output validated?).
We would ask that the reviewer clarify whether he means it is acceptable for 95% of the peptides or of the available primary amines to be labelled? It was our understanding that in SILAC experiments, which use the same precursor ion quantification as reductive dimethylation, there is a requirement for >95% incorporation of the heavy amino acid be demonstrated before mixing for quantification is performed (https://www.biochem.mpg.de/221777/SILAC). We do not see why this requirement should not be applied to chemical labelling and in this work we showed that the lowest proportion of unlabelled peptides was >5% and outside the range the reviewer specifies.
If the peptide is partially labelled, the chemistry of the peptide is being altered which will alter the retention time of the peptide. Zee and Garcia (Essays Biochem. 2012; 52: 147–163.) report (figure 3 of article) that monomethylation of a peptide causes a significant shift in retention time, 13 mins in the case of their gradient conditions. Yang et al (Anal Biochem. 2010 Jan 1; 396(1): 13–22.) report a retention time shift of one minute when a peptide is dimethylated. Because the algorithms used for precursor ion quantification require the light and heavy versions of the peptides to occur in the same MS1 scans, an unlabelled or partially labelled peptide at a different retention time will not contribute to the quantifiable abundance and this will not be known to the researcher. Therefore, we disagree with the reviewer that partially labelled peptides will be ‘perfectly usable for quantitative purposes’.
We don’t understand why the reviewer believes that the quoted sentence is an overstatement. It is simply a statement of fact, that we were able to analyse the prevalence of unlabelled peptides and did this using a script to extract the information from our search results.
It is thus important to use quantitative information next to database search results which has not been performed in this study. There are several possibilities available to the authors to explore the labelling efficiency of which a few are mentioned here. A first step would be to calculate the ratio between the labelled and unlabelled version(s) of each peptide sequence using non-normalized signal intensities. It is true that retention shifts due to the label or the label itself might influence the signal intensity but this is non-significant for dimethy labelling at a global level in our experience.
We tried to perform this analysis. Peaks Studio X can only perform Precursor Ion Quantification if a modification has been included in the initial database search. An unlabelled peptide has no modification to specify. We attempted to circumvent this by creating a modification called ‘un-dimethylated amine’ that was restricted to Lysine and added zero mass. This was of limited success and did not address the reviewer’s comment.
However, to address this problem during data analysis rather than before acquisition, the search algorithms and quantification software would have to be modified to look for and include unmodified and partially labelled peptides. We think that it would be better to improve the labelling efficiency than ask software developers to alter code to fix this problem.
Further, if incomplete labelling does interfere with quantitation this should be obvious from the relative quantitation results: the authors could test if the ratio between the labelled pairs statistically differ from the ratios of the other peptides from the same protein in these datasets, particularly if label swaps were performed. If the labelling efficiency is consistent between samples then the partially labelled peptides will (globally) show a similar distribution of respective ratio deviations as the complete labelled peptides and will not impact quantitative results. There are more alternative analyses feasible to investigate IF the labelling efficiency is indeed low in these datasets and IF the lower labelling efficiency impacts quantitative results (hence: increased variation should be substantially larger than the intra- and inter-assay variation; proper statistical analysis is warranted in any case).
We have attempted to do this by asking Peaks Studio X to identify unlabelled and partially labelled peptides. But our point above addresses the reviewer’s point here.
Minor concerns:
Figure 2: the authors show the best peptide-spectrum match (PSM) in a mascot search result from a (poor mass accuracy) dataset. The information shown in figure 2 does not proof the actual modification site by itself (particularly important for 2B). (1) provide delta scores for the subsequent lower (or equal scoring) peptide-spectrum matches for both spectra. (2) please provide the annotated MS/MS spectra with indicated diagnostic fragment ions to proof the site-specific localization of the dimethyl group.
General remark with respect to the database searches: N-terminal protein formylation might occur in specific species or organelles. We routinely detect N-terminal formylation of specific mitochondrial proteoforms in different organisms by top-down proteomics and subsequent shotgun proteomics. Dimethylation might not be discerned from formylation for peptides of Mr >2500 Da at <15ppm precursor mass tolerance with near perfect mass calibrated data files. However, if there is a few ppm mass offset in the acquired data formylated peptides can be readily detected as dimethylated peptides and vice versa. Did the authors consider this potential problem and performed variable PTM searches to check for potential issues? In addition, what were the exact protein sequence databases used in this study (source, version/date?)?
As mentioned above, it is likely that molecules of the same peptide that differ by formylation or demethylation will have different retention times allowing their differentiation. We agree with the reviewers comment that it is a potential issue but did not search all of the possible related modifications, nor did we look at mass error distributions because we were looking for a lack of labelling, not alternative PTMs that could further confound quantification. The databases used were the most up-to-date versions in UniProt at the time of analysis. This information has been added to the manuscript.
Line 201-203: “... the search parameters applied in these studies will either not assign all of the acquired spectra, resulting in a false negative, or assign a spectrum to the incorrect sequence resulting in a false positive”. Shotgun proteomics will never achieve 100% peptide-spectrum matches for a multitude of reasons (a.o. poor signal-to-noise spectra, poor fragmentation, chimeric spectra, sequence scrambling, unspecified combinations of modifications and amino acid variants, incorrect protein sequences in the database, statistics, etc etc).
While we agree with the reviewer’s comment, we do not see its relationship to the highlighted sentence.
Line 211-212: “In the case of these studies, the false negative rate is far greater than the false positive rate reported in the individual studies and this is of serious concern when quantitation is considered as it will lead to false values”. The false negative rate is always (!) higher than the false positive rate in decoy database search strategies for any given shotgun proteomics experiment. However, this by itself will not lead to false quantitative values per se IF the labelling efficiency is nearly identical between both samples. Moreover, the authors did not determine the false negative rate – this can by definition not be calculated for these datasets since it lacks ground truth information. It can be estimated by alternative statistical approaches that require a lot of mathematics and that take more than just missing dimethyl labels into account.
The reviewer is agreeing with the point that we are trying to make. In most circumstances, the false negative rate cannot be determined (How do you determine something that cannot be observed?) so it is not known if it is higher than the false positive rate. In this case, we can determine the false negative rate and this logic is described in the manuscript.
Line 229-245: This section highlights the unfortunate state of many datasets filed at online repositories and stresses the need to adhere to FAIR guidelines. However, in our experience authors that deposited datasets are often willing to provide additional information or data when asked to. Did the authors of this manuscript contact the owners of these datasets and asked for information / data files?
No, partly due to time restrictions and partly because we shouldn’t need to. That is the point we are trying to make, that data files submitted to public repositories should be correctly labelled when submitted.
Line 258-261: did the authors consider using different enzymes or chemical digestion methods to validate N-terminomics results / find missing N-termini? Why would N-terminal dimethylation solve the problem of “MS-unfriendly” N-terminal peptides? The dimethyl labelling confirms the N-terminus IF an N-terminal dimethylated peptide is detected but does not solve the problem of absent proof. A single enzyme will nearly certain not provide “MS-friendly” peptides for each N-terminus of every canonical protein sequence. As such, different digestion protocols can (or should) be used to fill missing data gaps and for cross-validation purposes. Moreover, calculations can be performed a priori to evaluate detectability of protein N-termini with the use of different digestion protocols. Moreover, it would be best to measure a sample before and after labelling of protein N-termini as PTM mass offsets in database searches can lead to ambiguous PTM identifications. In general, FDR calculations based on decoy database search strategies are just an estimate of the true false-discovery rate and individual incorrect PSMs will always occur even at extremely low FDR thresholds.
We haven’t done the experiments that generated the data that was analysed, so we had no control over the methods used. In our own work, we investigated using Lysarginase, but found that it cleaved non-specifically (anecdotally supported by other researchers), and Glu-C, which also did not produce many suitable peptides when applied to Mycoplasma.
Round 2
Reviewer 1 Report
I thank you authors for their efforts to clarify and address my comments. While they have answered my comments, I am not sure if this manuscript should be published as a full research article.
My recommendation would be that authors re-format and reconsider publishing it as a mini-review rather than a research article. I carefully read the whole manuscript, and repeatedly read the abstract. Unfortunately, I am not convinced this is full research article. The way introduction and result section is presented, I think its suitable for mini-review if additional literature information is added to it. While this will be up to the editors, my recommendation is to reconsider for a mini--review or other application notes rather than a research article.
Figures 1 and 2 are for educational purposes, and ha nothing to do with the research experiment.
Author Response
We thank the reviewer for taking the time to review our manuscript. The article has been changed to a Technical Note, which was not an option available to us during submission of the original manuscript.
Reviewer 3 Report
First of all I would like to thank the authors for their response. Unfortunately it seems that some of my remarks were misunderstood which I attempt to clarify below and hope that this helps the authors to improve the manuscript.
“We don’t understand why the reviewer believes that the quoted sentence is an overstatement. It is simply a statement of fact, that we were able to analyse the prevalence of unlabelled peptides and did this using a script to extract the information from our search results”. And “We do not see why this requirement should not be applied to chemical labelling and in this work we showed that the lowest proportion of unlabelled peptides was >5% and outside the range the reviewer specifies.”
I strongly disagree with both the original statement in the manuscript and the response. The qualitative information from search results (present/absent) can simply not be used to determine the labeling efficiency (which I tried to explain). It can be used to determine the presence of unlabeled peptides (and hence, indicate less than 100% labeling efficiency) but will not determine the actual labeling efficiency. A correct statement of fact would be “For 5.84-17.44% of the N-terminal labeled peptides incomplete labeling was detected through identification of their unlabeled counterparts via database searches”. Most likely, this is still an underestimate as e.g. unmodified versions of the partially labeled pairs escape detection due to MS/MS sensitivity limitations. Moreover, in multiplexed measurements the individual samples all contribute to the unlabeled fraction(s) of the N-terminal peptides which should be taken into account. Nevertheless, if the labeling efficiency is consistent between samples (which can be checked prior to mixing of the samples) they can be used for quantitation.
“... Therefore, we disagree with the reviewer that partially labelled peptides will be ‘perfectly usable for quantitative purposes’”. Indeed, they are not “perfectly” suitable. However, if the labelling efficiency is over >95% for a given peptide (which will still allow the unlabeled counterpart to be identified through DB searches) in two samples such peptides can be used for quanitative purposes as is typically the case in most shotgun proteomics studies.
“If the peptide is partially labelled... ...and this will not be known to the researcher”. I strongly agree with the authors that it is key to determine and report the labelling efficiency per sample prior to mixing and subsequent analysis. However, this should not be calculated from database search results. As the authors indicate later in their response most software does not report quantitative information for partially labelled pairs out of the box (I believe the authors refer to the PEAKS Q module here?) . If the peptide identification report in PEAKS studio v8.5 reports the precursor areas (like It does in PEAKS studio X?) then the quantitative information for partially labelled pairs is readily available and exported by the software. The script used by the authors could be modified to extract this information from the report and since the script already detected partially labelled pairs it should require very minimal scripting to determine the ratio between the unlabeled and labelled versions of the same peptide per sample. Of course, achieving 100% labeling efficiency would solve most issues but might be impossible to achieve and is not required per se.
“While we agree with the reviewer’s comment, we do not see its relationship to the highlighted sentence.” The sentence at line 201-203 in the manuscript is a fact statement that holds true for any shotgun proteomics experiment ( e.g. carbamidomethylation (Cys)). I understand that my remark was not clear perhaps which I will explain in more detail below: The authors indicate that false negative rate is far greater than the false positive rate in the individual studies and that this is of serious concern when quantitation is considered as it will lead to false values. This is true for nearly every shotgun proteomics study as the true false negative rate cannot be calculated. However, this might not be relevant for quantitation if the labelling efficiency for two given samples that are compared is identical (if a given peptide is 100fmol in both samples, and both samples have 97% labelling efficiency, then the ratio for this peptide between both samples is still 1 as 97fmol versus 97fmol of labelled products are measured, respectively). This section in the results part of the manuscript is a statement of common knowledge but the authors should analyze and report if the labelling efficiency in these studies was indeed a problem and if this has lead to significantly incorrect quantitative values. E.g. determine if the partially labeled peptide pairs corresponded with large relative ratios that are outside the ratio distribution of all labeled peptides in the experiment, etc. No results are shown to evaluate if false quantitative values were produced by these studies.
“The reviewer is agreeing with the point that we are trying to make. In most circumstances, the false negative rate cannot be determined (How do you determine something that cannot be observed?) so it is not known if it is higher than the false positive rate. In this case, we can determine the false negative rate and this logic is described in the manuscript.” Please see my remark above with respect to labeling efficiency. The current state of the manuscript can be summarized as follows: (i) unlabeled counterparts of labelled peptide pairs could be identified through database searches for different N-terminomics datasets which (ii) could potentially interfere with correct relative quantitation in the analyzed studies. (i): No labelling efficiency is provided within the manuscript, just presence of unlabeled peptides that indicate less than 100% labelling efficiency (e.g. efficiency could still be 99.9%). (ii): there is no evidence, nor is any analysis performed, to show if indeed the labelling efficiency was too low and/or irreproducible for individual samples in respective studies to have any significant effect on the quantitative output.
Finally, I would like to ask the authors respond to my concern with respect to figure 2.
Author Response
We thank the reviewer for the considerable time they have taken to review our manuscript with great rigor. Their comments and suggestions have improved the manuscript. Our responses to the comments are in red text.
First of all I would like to thank the authors for their response. Unfortunately it seems that some of my remarks were misunderstood which I attempt to clarify below and hope that this helps the authors to improve the manuscript.
“We don’t understand why the reviewer believes that the quoted sentence is an overstatement. It is simply a statement of fact, that we were able to analyse the prevalence of unlabelled peptides and did this using a script to extract the information from our search results”. And “We do not see why this requirement should not be applied to chemical labelling and in this work we showed that the lowest proportion of unlabelled peptides was >5% and outside the range the reviewer specifies.”
I strongly disagree with both the original statement in the manuscript and the response. The qualitative information from search results (present/absent) can simply not be used to determine the labeling efficiency (which I tried to explain). It can be used to determine the presence of unlabeled peptides (and hence, indicate less than 100% labeling efficiency) but will not determine the actual labeling efficiency. A correct statement of fact would be “For 5.84-17.44% of the N-terminal labeled peptides incomplete labeling was detected through identification of their unlabeled counterparts via database searches”. Most likely, this is still an underestimate as e.g. unmodified versions of the partially labeled pairs escape detection due to MS/MS sensitivity limitations. Moreover, in multiplexed measurements the individual samples all contribute to the unlabeled fraction(s) of the N-terminal peptides which should be taken into account. Nevertheless, if the labeling efficiency is consistent between samples (which can be checked prior to mixing of the samples) they can be used for quantitation.
It is our belief that the reviewer is in agreement with what we are trying to convey, but is expressing it in slightly different terms and we will re-word the manuscript accordingly. We have removed all references to labelling ‘efficiency’ and replaced it with more specific terms.
The reviewer further supports the point we are trying to make by writing ‘if the labeling efficiency is consistent between samples (which can be checked prior to mixing of the samples) they can be used for quantitation’. This statement means that researchers must search their data with dimethylation set as a variable modification so that the number of unlabelled primary amines is known. The researcher can then attempt to further label the unlabelled amines or report that not all amines are labelled. This can be done on single samples prior to mixing or after mixing and if the labelling is deemed sufficiently complete, then quantitation can be performed. The manuscript states ‘we suggest that the extent of this should be empirically analysed and reported’ and we have added to this statement at the reviewer’s suggestion.
The percentages stated in the manuscript refer to the re-analysed data, NOT the original data located in public repositories. For each dataset, we determined the total number of peptides that could be identified by setting variable modifications which results in a greater number of peptides being identified than in the original reports. The number of peptides containing a primary amine that has not been dimethylated are then counted and expressed as a percentage of the total peptides identified.
“... Therefore, we disagree with the reviewer that partially labelled peptides will be ‘perfectly usable for quantitative purposes’”. Indeed, they are not “perfectly” suitable. However, if the labelling efficiency is over >95% for a given peptide (which will still allow the unlabeled counterpart to be identified through DB searches) in two samples such peptides can be used for quanitative purposes as is typically the case in most shotgun proteomics studies.
We agree with the reviewer’s clarification. We have amended the manuscript to reflect the comment.
“If the peptide is partially labelled... ...and this will not be known to the researcher”. I strongly agree with the authors that it is key to determine and report the labelling efficiency per sample prior to mixing and subsequent analysis. However, this should not be calculated from database search results. As the authors indicate later in their response most software does not report quantitative information for partially labelled pairs out of the box (I believe the authors refer to the PEAKS Q module here?) . If the peptide identification report in PEAKS studio v8.5 reports the precursor areas (like It does in PEAKS studio X?) then the quantitative information for partially labelled pairs is readily available and exported by the software. The script used by the authors could be modified to extract this information from the report and since the script already detected partially labelled pairs it should require very minimal scripting to determine the ratio between the unlabeled and labelled versions of the same peptide per sample. Of course, achieving 100% labeling efficiency would solve most issues but might be impossible to achieve and is not required per se.
The reviewer is correct that the data can be ‘corrected’ post acquisition in the way described. However, we more firmly agree with the reviewer’s later point that ensuring complete labelling prior to performing the quantitative experiment is a better solution. The data correction suggested is likely to lead to more questions because we suspect that the completeness of labelling will vary with different peptides, leading to more data manipulation to correct.
“While we agree with the reviewer’s comment, we do not see its relationship to the highlighted sentence.” The sentence at line 201-203 in the manuscript is a fact statement that holds true for any shotgun proteomics experiment ( e.g. carbamidomethylation (Cys)). I understand that my remark was not clear perhaps which I will explain in more detail below: The authors indicate that false negative rate is far greater than the false positive rate in the individual studies and that this is of serious concern when quantitation is considered as it will lead to false values. This is true for nearly every shotgun proteomics study as the true false negative rate cannot be calculated. However, this might not be relevant for quantitation if the labelling efficiency for two given samples that are compared is identical (if a given peptide is 100fmol in both samples, and both samples have 97% labelling efficiency, then the ratio for this peptide between both samples is still 1 as 97fmol versus 97fmol of labelled products are measured, respectively). This section in the results part of the manuscript is a statement of common knowledge but the authors should analyze and report if the labelling efficiency in these studies was indeed a problem and if this has lead to significantly incorrect quantitative values. E.g. determine if the partially labeled peptide pairs corresponded with large relative ratios that are outside the ratio distribution of all labeled peptides in the experiment, etc. No results are shown to evaluate if false quantitative values were produced by these studies.
We agree with the reviewer’s comments but our intention is to point out that incomplete labelling is potentially a problem, NOT to reanalyse and reinterpret the studies of others. Unfortunately, we do not have the personnel resources available to perform the requested analysis in a short timeframe.
“The reviewer is agreeing with the point that we are trying to make. In most circumstances, the false negative rate cannot be determined (How do you determine something that cannot be observed?) so it is not known if it is higher than the false positive rate. In this case, we can determine the false negative rate and this logic is described in the manuscript.” Please see my remark above with respect to labeling efficiency. The current state of the manuscript can be summarized as follows: (i) unlabeled counterparts of labelled peptide pairs could be identified through database searches for different N-terminomics datasets which (ii) could potentially interfere with correct relative quantitation in the analyzed studies. (i): No labelling efficiency is provided within the manuscript, just presence of unlabeled peptides that indicate less than 100% labelling efficiency (e.g. efficiency could still be 99.9%). (ii): there is no evidence, nor is any analysis performed, to show if indeed the labelling efficiency was too low and/or irreproducible for individual samples in respective studies to have any significant effect on the quantitative output.
The manuscript has been altered to remove references to ‘efficiency’ and replace them with more appropriate terminology. As stated above, our intention is not to reinterpret other people’s published conclusions. Our intention was to draw attention to a problem that can be easily avoided through the use of correct search settings.
Finally, I would like to ask the authors respond to my concern with respect to figure 2.
Figure 2 was altered in the last version of the manuscript. Track Changes allows the deleted figure to be seen to show that it was deleted. We have removed the old figure for clarity.